# The Role of Circular RNA for Early Diagnosis and Improved Management of Patients with Cardiovascular Diseases

**DOI:** 10.3390/ijms25052986

**Published:** 2024-03-04

**Authors:** Claudia Alexandrina Goina, Daniela Marcela Goina, Simona Sorina Farcas, Nicoleta Ioana Andreescu

**Affiliations:** 1Doctoral School, Discipline of Genetics, “Victor Babes” University of Medicine and Pharmacy, Piata Eftimie Murgu 2, 300041 Timisoara, Romania; claudia.goina@gmail.com; 2Faculty of Animal Husbandry and Biotechnologies, University of Agricultural Sciences and Veterinary Medicine of Banat, Calea Aradului 119, 300645 Timisoara, Romania; danmarce1@yahoo.com; 3Department of Microscopic Morphology, Discipline of Genetics, Genomic Medicine Centre, “Victor Babeș” University of Medicine and Pharmacy, Piata Eftimie Murgu 2, 300041 Timisoara, Romania; nicollandreescu@yahoo.com

**Keywords:** circRNA, cardiovascular disease, regulatory molecules, biological processes

## Abstract

Cardiovascular diseases (CVDs) are responsible for approximately 17.9 million deaths every year. There is growing evidence that circular RNAs (circRNAs) may play a significant role in the early diagnosis and treatment of cardiovascular diseases. As regulatory molecules, circular RNAs regulate gene expression, interact with proteins and miRNAs, and are translated into proteins that play a key role in a wide variety of biological processes, including the division and proliferation of cells, as well as the growth and development of individuals. An overview of the properties, expression profiles, classification, and functions of circRNAs is presented here, along with an explanation of their implications in cardiovascular diseases including heart failure, hypertension, ischemia/reperfusion injury, myocardial infarction, cardiomyopathies, atherosclerosis, and arrhythmia.

## 1. Introduction

Cardiovascular diseases (CVDs) are the leading global cause of death, accounting for 17.9 million fatalities annually. These diseases include heart and blood vessel disorders, and approximately four out of five CVD deaths are due to heart attacks and strokes, with one-third occurring before the age of 70 [1].

Cardiac enzymes, utilized since the mid-20th century for myocardial infarction assessment, have seen advancements with more precise biomarkers in recent years. Troponins, the most well-known and significant cardiac proteins in contemporary medicine, aid in diagnosing acute myocardial ischemia [2]. However, their utility for early heart disease detection is limited due to rapid fluctuations influenced by various factors such as illnesses, medications, genetics, or age [3]. Recent developments have introduced more accurate biomarkers, replacing the older ones.

In recent years, circular RNAs have gained increasing importance as a research topic in diabetes, neoplasia, and cardiovascular disease. Further investigation is needed to understand their potential impact on pathogenic processes in the cardiovascular system.

Circular RNA molecules (circRNAs) are covalently closed single-stranded RNA molecules found in many eukaryotic cells. As regulatory molecules, circular RNAs play a crucial role in regulating various biological processes, including gene expression control, interaction with proteins and miRNAs, and potential translation into proteins. These functions are vital for cell division, proliferation, growth, and development [4].

It will take considerable effort to develop novel approaches to cardiovascular disease diagnosis and treatment to understand how this knowledge might lead to novel approaches.

Collaboration between molecular biologists, cardiologists, and bioinformaticians is essential for advancing circRNA research. Molecular biologists can provide insights into the biogenesis and regulation of circRNA, while cardiologists can offer clinical perspectives by linking circRNA findings to cardiovascular conditions. Bioinformaticians analyze vast datasets to extract meaningful interpretations. This interdisciplinary approach accelerates circRNA research and translates discoveries into practical applications for improved cardiovascular care.

## 2. Uncovering CircRNAs

Circular RNAs (circRNAs) were discovered in 1976 when Sanger et al. identified viruses with single-stranded, covalently closed circular RNA molecules [5]. Presently, 32,000 circRNAs have been identified [6] and are recognized as non-coding RNAs, potentially serving as therapeutic targets for various diseases, including cardiovascular diseases [7].

## 3. Classification of CircRNAs

Circular RNAs fall into four categories based on their genomic locations [8,9] (Table 1):(1)Exonic circular RNAs (ecircRNAs) were found to be processed from exons and are located in the center of RefSeq genes, with potential functions in transcription and gene proliferation [10];(2)Circular intronic RNAs (ciRNAs) are found mainly in the nucleus, aiding gene transcription and interacting with RNA polymerase II [11];(3)Exon–intron circular RNAs (EIciRNAs) are found in the nucleus and are associated with U1 snRNPs, initiating gene transcription [12];(4)tRNA intronic circular RNAs (tricRNAs) are found mainly in the cytoplasm, offering potential in RNA-based applications with relevance to neurodegenerative diseases linked to tRNA processing factors [9,13].

**Table 1 ijms-25-02986-t001:** An overview of the classification of circular RNAs. This table describes distinct types, locations, joint sites, and functions of circular RNAs. EcircRNA—exonic.

Name	Type	Location	Joint Site	Function	Reference
EcircRNA	Exonic	Cytoplasm	3′-5′ phosphodiester bond	miRNA sponges, interaction with RBP (RNA binding protein), translation	[10]
CiRNA	Intronic	Nucleus	2′-5′ phosphodiester bond	Gene transcription	[11]
EIciRNA	Exon-intron	Nucleus	3′-5′ phosphodiester bond	Gene transcription	[12]
TricRNA	tRNA intronic	Cytoplasm	3′-5′ phosphodiester bond	Unknown	[9,13]

## 4. Functions of CircRNAs (Figure 1)

CircRNA, a vital part of biological development, plays a significant role in gene expression regulation and supplies valuable information about disease diagnosis. Evidence shows that nuclear circRNAs not exported from the nucleus may affect transcriptional regulation.

**Figure 1 ijms-25-02986-f001:**
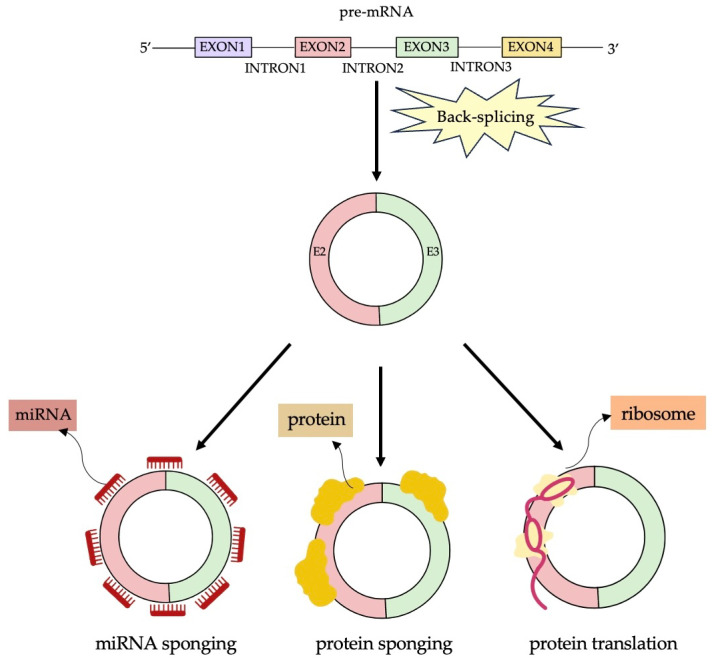
The biogenesis of circular RNAs and their functions (CircRNAs act as miRNA sponges, RNA-binding protein, and translators in protein synthesis). Created in Notability app.

### 4.1. Circular RNAs Can Regulate Splicing and Transcription

Some EIciRNAs can be inhibited to reduce the transcription of the corresponding genes. In order to enhance gene expression, EIciRNAs directly interact with U1snRNPs (U1 small nuclear ribonucleoproteins), which then bind to Pol II at the gene promoters [12].

### 4.2. Circular RNAs Act as miRNA Sponges

mRNAs with complementary miRNAs are negatively regulated through partial base pairing with their UTRs. Through computational analysis, miRNA binding sites have been found in circRNAs. “Sponging” occurs due to complementary circRNA and miRNA sequences interacting to bind and hold miRNAs together [14].

In gene expression regulation, abundant endogenous circular RNA (circRNA) molecules act as effective sponges for microRNAs (miRNAs) (Figure 1) and expand the range of regulatory functions. Circular RNA molecules show intrinsic resistance to exonucleolytic degradation, contributing to their effectiveness in regulating gene expression [15,16].

### 4.3. Circular RNAs Act as RNA-Binding Protein

It is known that RNA-binding proteins (RBPs) play a role in a wide range of cellular processes, including proliferation, differentiation, transmigration, apoptosis, senescence, and response to oxidative stress. Therefore, they take part in the post-transcriptional regulation of RNA, including activities such as splicing, transport, and translation [17].

There has been evidence that specific circular RNAs (circRNAs) interact with other RNAs to block cell division at the G1/S phase.

It has been proven that specific circular RNAs (circRNAs) can change the stability of messenger RNAs (mRNAs), such as CDR1as, which bind to mRNA and form a durable duplex [18].

There is substantial evidence that the tertiary structure of RNA molecules influences the interaction between circRNAs and RBPs (Figure 1). While bioinformatic analysis has not revealed much enrichment of RBP-binding sites in circRNA sequences, circRNAs are still expected to interact with RBPs through specific binding sites. As a result, there may be other factors that contribute to the facilitation of these interactions as well [19].

### 4.4. Circular RNAs Act as Translators in Protein Synthesis

At first, it was believed that circRNAs were not capable of being translated. Despite this, circRNAs are predominantly found in the cytoplasm, derived from exons that encode proteins, which has led to the question of whether circRNAs can be attached to ribosomes and undergo protein translation (Figure 1).

In vivo and in vitro studies have shown that the 40S subunit of eukaryotic ribosome can attach to a modified circular RNA with a single IRES sequence, thus initiating translation [20,21]. There is evidence that Circ-ZNF609, which carries IRES, can translate proteins [22]. In recent studies, MBL3 was proven to have the ability to produce mutations in proteins within the head of flies [23]. An additional study showed that, even without any specialized component facilitating ribosome entry into the cell, the artificial circRNA carrying many FLAG coding sequences can translate proteins through rolling circle amplification (RCA) [24]. It has also been shown that the most common base modification found in RNA, known as N6, eases the translation of proteins from circulating ribonucleases into peptides in human cells [25].

## 5. Circular RNA in Cardiovascular Disease

It was revealed using deep sequencing of cardiac tissue that there is an abundance, evolution, conservation, and differentiation of circRNA expressed in the heart depending on the developmental stage (Figure 2). The role of circular RNAs in cardiovascular diseases has not been proven definitively, but recent research suggests that several circRNAs act as competing endogenous RNAs that regulate gene expression. Evidence shows that circRNAs are generated by several genes associated with cardiovascular disease, including Ryr2, Ttn, and Dmd [26].

Regarding the clinical implications, circRNA research holds great promise for personalized medicine and therapeutic interventions in cardiovascular health. Circular RNAs play a variety of regulatory functions that represent potential targets for developing innovative treatments for conditions such as cardiovascular diseases. As the expression of circRNAs varies significantly among individuals, personalized medicine can provide tailored treatment plans and improve patient outcomes. CircRNAs also hold promise as biomarkers, offering non-invasive methods for disease monitoring and timely interventions. Therefore, circRNA research has the potential to revolutionize cardiovascular medicine through targeted therapies and improved diagnostic accuracy.

### 5.1. Congestive Heart Failure

Congestive heart failure (CHF) is a complex clinical syndrome caused by structural or functional changes in the heart that impair the ability of the ventricles to fill or empty blood. Since no definitive test is available for diagnosing heart failure, ancillary testing such as chest X-rays, electrocardiograms, and echocardiograms are used in conjunction with a thorough examination of the patient [27].

Diastolic dysfunction and systolic dysfunction result in a reduced cardiac output and heart failure. The most common causes of systolic dysfunction (defined as less than 50% left ventricular ejection fraction) are heart disease, idiopathic dilated cardiomyopathy, hypertension, and valvular heart disease. A diastolic dysfunction (the presence of reduced left ventricular filling but preserved systolic function) is common in 40–50% of patients with heart failure. It is more prevalent in women, and it increases with age. Diastolic dysfunction can be caused by many of the same conditions that cause systolic dysfunction. Several factors contribute to heart failure, including hypertension, ischemic heart disease, hypertrophic cardiomyopathy, and restrictive cardiomyopathy [27,28].

By analyzing circulating RNAs, enriched and stable in whole blood, platelets, and plasma, it may be possible to predict and stratify the incidence of heart failure-related death. Several circRNAs have been validated using quantitative real-time PCR in plasmas of patients with heart failure (hsa_circ_0112085, hsa_circ_0062960, hsa_circ_0011464, hsa_circ_0053919, and hsa_circ_0014010) and 696 differentially expressed circRNAs have been found. Among the heart failure biomarkers, hsa_circ_0062960 was the standout candidate for significant differences in expression and diagnostic utility [29].

It is well known that heart failure is associated with venous thromboembolism, stroke, and sudden death on a global basis. Several factors play a role in the development of HF, including platelet activity [30]. An essential feature of the clinical presentation of patients with HF is an increase in platelet-derived adhesion molecules and a higher level of whole-blood aggregation [31,32].

Through its ancestry, Hsa_circ_0069197 is related to the WDR1 gene family, which has been proven to suppress platelet activity. Overexpression of WDR1 in cardiovascular diseases is expected to lead to platelet-mediated pathogenesis. Additionally, hsa_circ_0062960, an overly sensitive biomarker, is derived from a gene called DEPDC5, which helps to properly develop the heart, blood vessels, and lymphatic vessels [33,34].

### 5.2. Hypertension

Essential hypertension (EH) refers to high blood pressure caused by an unknown cause, which has been associated with an increased risk of brain, heart, and kidney damage. Over 90% of people in industrialized countries are believed to be at risk of becoming hypertensive (blood pressure levels over 140/90 mm Hg) at some point in their lives. Several cardiovascular risk factors are usually associated with essential hypertension, including aging, obesity, insulin resistance, diabetes, and hyperlipidemia. As hypertensive cardiovascular disease progresses, subtle damage may occur to target organs, such as left ventricular hypertrophy, microalbuminuria, and cognitive impairment. Most catastrophic events, such as strokes, heart attacks, renal failure, and dementia, are caused by uncontrolled hypertension. The prevention of long-term irreversible damage associated with hypertension requires early diagnosis and treatment [35].

The role of circRNAs in cardiovascular diseases is not proven definitively, but recent research suggests that several circRNAs act as competing endogenous RNAs that regulate gene expression.

Circ_0126991 has been identified as a critical player in essential hypertension (EH) (Table 2). This circular RNA functions as a sponge for miR-10a-5p [36]. Another noteworthy CircRNA, Hsa_circ_0037897, has been demonstrated to be upregulated in EH and acts as a sponge for hsamiR-145-5p (Table 2) [37]. Hsa_circ_0037911 and Hsa-miR-637 have been shown to exhibit elevated levels in essential hypertensive individuals, suggesting that they may serve as valuable early-stage EH biomarkers (Table 2) [38]. Additionally, it has been demonstrated that Hsa_circ_0105015, when hyperexpressed, increases EH risk and is associated with inflammatory pathways. Its combination with hypoexpression of hsa-miR-637 may be an early indicator of EH (Table 2) [39].

### 5.3. Ischemia/Reperfusion (I/R) Injury and Myocardial Infarction (MI)

“Ischemia and reperfusion” refers to the pathological condition of restricted blood supply to an organ followed by a reorganization of the blood flow and reoxygenation. The classic manifestation of this type of occlusion is an embolus that obstructs the arterial blood supply, resulting in a metabolic imbalance and hypoxia of the tissues. As a result of the restoration of blood flow and reoxygenation, tissue damage is often exaggerated in combination with a profound inflammatory response [40].

Heart attacks caused by myocardial infarction occur because of plaque forming in the arteries, resulting in decreased blood flow to the heart and injury to the heart muscles due to inadequate oxygen supply [41].

Increasing evidence suggests that circRNAs may be involved in cardiac ischemia-reperfusion injury and myocardial infarction (MI) caused by miRNA interactions (Table 3) (Figure 3a,b). For example, CDR1as (CiRS-7) has been demonstrated to be upregulated in cardiac infarcts, contributing to the pathogenesis of myocardial infarction (MI) [15,16,18,42]. CircNFIX, regulated by a super-enhancer, is overexpressed in MI, impacting cardiomyocyte proliferation [43]. Conversely, MICRA has been revealed to exhibit decreased levels in the peripheral blood of MI patients and is associated with left ventricular dysfunction [44,45]. CircPostn has been found to be elevated in MI patients, actively participating in myocardial injury and contributing to cardiac remodeling. At the same time, CircPAN3 has been associated with cardiac fibrosis in MI through the miR-221/FoxO3/ATG7 axis [46,47]. Additionally, other CircRNAs, such as CircFASTKD1, CircHIPK3, CircROBO2, MFACR, CircNNT, CircSRY, CircRNA_081881, Circ_0060745, and Circ_010567, play roles in various aspects of MI, including angiogenesis, regeneration, apoptosis, and fibrosis [48,49,50,51,52,53,54,55,56,57,58,59,60,61,62].

### 5.4. Dilated Cardiomyopathy (DCM) and Hypertrophic Cardiomyopathy (HCM)

Cardiomyopathy is a condition affecting the heart muscle characterized by abnormal chamber and wall sizes or functional contractile abnormalities. This condition is characterized by systolic or diastolic dysfunction without coronary artery disease, hypertension, valvular disease, or congenital defects in the heart [63].

Cardiomyopathies can be classified as primary or secondary. “Primary cardiomyopathy” refers to conditions that exclusively or primarily affect the heart muscle. These conditions can be inherited, non-inherited, or acquired. Secondary cardiomyopathies are caused by systemic or multiorgan diseases that harm the heart [64].

Dilated cardiomyopathy is one of the most common cardiomyopathies today, and a variety of factors can cause it. This disorder causes the left or both ventricles to dilate and become incapable of contracting [65]. Besides causing sudden cardiac death, dilated cardiomyopathy can lead to heart failure, which has a high cost burden due to the high rate of hospitalizations and the potential need for heart transplantation [65].

Although hypertrophic cardiomyopathy is the most common cause of sudden death among young people (including athletes) and can lead to functional disability as a result of heart failure and stroke, most individuals are probably undiagnosed and do not experience significant reductions in life expectancy or substantial symptoms. The clinical diagnosis of left ventricular hypertrophy is based on echocardiography or cardiovascular magnetic resonance imaging [66].

The disease has been associated with more than 1400 genetic mutations in 11 or more cardiac sarcomere genes [66]. It is believed that most cases are caused by mutations in genes encoding sarcomeric proteins, such as beta-myosin heavy chain, myosin-binding protein C, and troponin I and T (autosomal dominant). The disease can also be caused by other genetic disorders, such as Friedreich’s ataxia, amyloidosis, and mitochondrial disease.

Researchers have proved that circRNAs derived from the TTN gene are differentially expressed in neonatal and adult rat hearts, suggesting that circRNAs play an essential role in cardiac development [67]. The expression of circRNAs is altered during the differentiation of cardiac progenitors into cardiomyocytes, suggesting that circRNAs are involved in the specification of cardiac cells, which appears to be a universal phenomenon [68].

Additionally, circular RNAs play a crucial role cardiac dysfunction and development. For example, heart-related circRNA (HRCR) inhibits miR-233 activity by acting as an endogenous sponge for miRNA-223, resulting in cardiac hypertrophy and heart failure [58].

In the context of dilated cardiomyopathy (DCM), numerous circRNAs have been implicated, demonstrating altered expression profiles in affected patients (Table 4). Remarkable instances encompass CircSLC8A1, CircCHD7, and CircATXN10, originating from the Titin gene, SCAF8, TIAM2, and the Qki gene. Importantly, it has been demonstrated that these circRNAs, associated with mutations, serve as potential sponges for miRNAs, playing pivotal roles in doxorubicin-induced cardiotoxicity [69,70,71].

Pathological myocardial hypertrophy, which is one of the leading causes of morbidity and mortality in developed countries and regions, is known to lead to heart failure, arrhythmias, and even sudden cardiac death [72]. CircRNAs can serve as new diagnostic or therapeutic targets for diseases related to myocardial hypertrophy due to their potential to affect pathological processes (Table 5).

### 5.5. Atherosclerosis

Atherosclerosis, which affects the arterial wall, is common among major conduit arteries. As a result of lipid retention, oxidation, and modification, chronic inflammation may ultimately lead to thrombosis or stenosis. In the context of atherosclerosis, there can be detrimental stenosis accompanied by distal ischemia, which may lead to death. Additionally, such lesions may result in the thrombotic occlusion of major vessels that supply the heart, brain, legs, and other organs. Initially, this disease affects the intima or inner lining of an artery, and then gradually spreads to the media and adventitia. Inflammation and low-density lipoproteins (LDLs) may both contribute to atherosclerosis. Among the most significant risk factors are hypertension, smoking, diabetes, obesity, and genetic predisposition [74].

With advances in deep sequencing techniques and data analysis methods, numerous studies have examined the role of circRNAs in atherosclerosis progression. There have even been suggestions that some circRNAs may be used as biomarkers for atherosclerosis (Table 6) [75].

Recent studies have shown that in individuals with atherosclerosis, CircRNA_102541 is overexpressed, marking it as a potential therapeutic target [76]. Similarly, CircRNA-0044073 exhibits an upregulation in atherosclerotic blood cells, fostering cell proliferation and targeting miR-107 [77]. CircRNA-0124644 has been found in peripheral blood samples as a biomarker for coronary artery disease, with a sensitivity and a specificity of 0.861 and 0.626, respectively, suggesting it is a biomarker for coronary artery disease [78]. These findings emphasize the intricate regulatory roles of circRNAs in cardiovascular diseases, offering valuable insights for potential diagnostic and therapeutic strategies.

## 6. Arrhythmia

Cardiac arrhythmias represent irregularities in the normal activation or beating of the heart’s myocardium. The sinus node initiates a depolarization wave, progressing through the atrium, atrioventricular (AV) node, and His-Purkinje system, systematically depolarizing the ventricle. The presence or absence of structural heart disease influences the severity of these irregularities. Conditions such as atrial fibrillation (AF) are typically considered benign. However, in individuals with coronary heart disease or severe left ventricular dysfunction, arrhythmias may lead to heart failure or sudden cardiac death [79].

Atrial fibrillation (AF) is an irregular heart rhythm distinguished by the rapid and erratic beating of the atria. A recent study highlighted the significant role of circRNAs in the development and progression of AF, identifying specific circRNAs that are differentially expressed in both the left and right atrial appendages of AF patients (Table 7). The research suggested that circRNAs, particularly hsa_circ_0003965, may be involved in AF through a unique regulatory mechanism potentially associated with the glucagon signaling pathway [80].

Additionally, a ceRNA network analysis conducted by researchers implicated specific circRNAs, such as hsa_circ_0000075 and hsa_circ_0082096, in the pathogenesis of AF through the TGF-beta signaling pathway. These findings contribute to a deeper understanding of the molecular mechanisms underlying AF, particularly the association with TGF-beta signaling and atrial fibrosis [80].

In another study, attention was directed toward circCAMTA1, a circRNA linked to AF. Elevated circCAMTA1 levels were noted in atrial tissues of AF patients and angiotensin-II-treated atrial fibroblasts. Its expression exhibited a correlation with collagen and α-SMA levels in AF patients. In vitro experiments demonstrated that circCAMTA1 knockdown impeded fibroblast proliferation and diminished the expression of genes associated with atrial fibrosis, while overexpression yielded contrasting effects. In vivo, circCAMTA1 knockdown alleviated atrial fibrosis, reducing AF incidence, duration, and collagen synthesis. Functionally, circCAMTA1 facilitated atrial fibrosis by suppressing miR-214-3p’s inhibitory effect on transforming growth factor β receptor 1 (TGFBR1) expression. This evidence highlights the circCAMTA1/miR-214-3p/TGFBR1 axis as a potential therapeutic target for AF [81].

## 7. Discussion

In recent years, circular RNAs have gained increasing importance as a research topic in diabetes, neoplasia, and cardiovascular disease. CircRNA, a vital part of biological development, plays a significant role in gene expression regulation and provides valuable information regarding disease diagnosis.

This paper briefly summarized the current understanding of circRNA properties, expression profiles, classification, and functions of circRNAs, and provided an explanation of their implications in cardiovascular diseases.

The evidence suggests that different circular RNAs can increase infarct size, promote cell death and apoptosis, suppress cardiac function, promote or suppress cardiac regeneration, disrupt protein structure and function, promote pyroptosis and aggravate I/R injury, and promote the proliferation of the vascular wall and progression to atherosclerosis.

Recent research indicates that several circRNAs act as competing endogenous RNAs that regulate gene expression. Evidence shows that circRNAs are generated by several genes associated with cardiovascular disease, including Ryr2, Ttn, and Dmd [26]. Specific circRNAs have been identified as potential biomarkers for various cardiovascular conditions.

Several circRNAs have been validated using quantitative real-time PCR in plasmas of patients with heart failure (hsa_circ_0112085, hsa_circ_0062960, hsa_circ_0011464, hsa_circ_0053919, and hsa_circ_0014010) and 696 differentially expressed circRNAs have been found. Among the heart failure biomarkers, hsa_circ_0062960 was the standout candidate for significant differences in expression and diagnostic utility [29].

Research has revealed a close association between essential hypertension and circular RNAs (circRNAs), with circ_0126991 and hsa_circ_0037897 emerging as noteworthy candidates. The intricate interplay observed between circ_0126991 and miR-10a-5p, as well as hsa_circ_0037897 and hsamiR-145-5p, strongly suggests their active involvement in the regulatory mechanisms underlying essential hypertension. These circRNAs are likely key players in the complex molecular landscape contributing to hypertension. Additionally, identifying various circRNAs, particularly hsa_circ_0037911, positions them as potential early-stage biomarkers for essential hypertension, offering promising prospects for diagnostic applications and emphasizing their role in comprehending and addressing this medical condition [36,37,38,39].

Several circRNAs have been identified to be implicated in the pathogenesis of myocardial infarction (MI), where CDR1as stands out as a significant circRNA functioning as a sponge or inhibitor of miR-7a/b [15,16,18]. Regulated by a super-enhancer, CircNFIX affects cardiomyocyte proliferation and angiogenesis post-MI [43]. The reduction in MICRA levels in the peripheral blood of MI patients suggests its potential as an indicator of left ventricular dysfunction [44,45].

Various circRNAs, including CircPostn, CircPAN3, CircFASTKD1, CircHIPK3, CircROBO2, MFACR, Circ NNT, Circ SRY, CircRNA_081881, Circ_0060745, and Circ_010567, have been shown to play roles in diverse aspects of myocardial infarction (MI) pathophysiology. CircPostn demonstrated heightened levels in MI patients, exacerbating myocardial injury and contributing to cardiac remodeling. Conversely, the decrease in CircFASTKD1 was associated with improved cardiac function and enhanced repair post-MI. Circ HIPK3 facilitated cardiac regeneration by stimulating cardiomyocyte mitosis and reducing scar size [46,47,48,49,50,51,52,53,54,55,56,57,58,59,60,61,62].

The interaction between CircROBO2 and miR-1184 enhanced myocardial viability, suppressed apoptosis, and indicated its potential as a biomarker for acute myocardial infarction [52,53]. MFACR, Circ NNT, Circ SRY, CircRNA_081881, Circ_0060745, and Circ_010567 have been demonstrated to play crucial roles in the progression of myocardial infarction (MI), constituting potential targets for precision diagnosis and treatment [54,55,56,57,58,59,60,61,62]. Further research is essential to fully understand the intricate relationships between these circRNAs, miRNAs, and their regulatory mechanisms in MI pathophysiology.

Scientists have proved that circRNAs derived from the TTN gene are differentially expressed in neonatal and adult rat hearts, suggesting that circRNAs play an essential role in cardiac development [67].

Additionally, circular RNAs play a crucial role in cardiac dysfunction and development. For example, heart-related circRNA (HRCR) inhibits miR-233 activity by acting as an endogenous sponge for miRNA-223, resulting in cardiac hypertrophy and heart failure [58].

The expression of circRNAs is altered during the differentiation of cardiac progenitors into cardiomyocytes, suggesting that circRNAs are involved in the specification of cardiac cells, which appears to be a universal phenomenon [68].

In dilated cardiomyopathy (DCM), various circRNAs, including Circ ALMS1_6, CircTTN_34, CircTTN_52, CircTTN_70, CircTTN_132, CircSCAF8_e4, TIAM2_e2, circRYR2_71, circRYR2_95, circDNAJ6C, circSLC8A1, circCHD7, and circATXN10, demonstrated altered expression, as supported by scientific research [69,70,71]. Circ ALMS1_6 and CircTTN notably exhibited a downregulated expression, impacting cardiac remodeling. In contrast, circRYR2 and circDNAJ6C displayed downregulation, affecting protein structure and function [69,70,71].

Concerning hypertrophic cardiomyopathy (HCM), specific circRNAs such as Circ TMEM56, Circ DNAJC6, and CircMBOAT2 have been identified with downregulated expression in individuals with HCM. This downregulation implies the potential utility of these circRNAs as quantitative indicators of disease severity in the context of HCM [73].

Recent studies have explored the role of circular RNAs (circRNAs) in atherosclerosis, revealing key players such as CircRNA_102541 and PLK1, which exhibit overexpression, while miR-296-5p levels were reduced in atherosclerosis specimens [76]. CircRNA-0044073, upregulated in atherosclerotic blood cells, acted as an miR-107 sponge, impacting vascular cell proliferation and activating the JAK/STAT pathway [77]. Higher circR284 levels than miR-221 in carotid plaques suggested biomarker potential [78]. CircRNA-0124644 in peripheral blood served as a biomarker for coronary artery disease [78]. These findings underscored circRNAs’ diverse roles in atherosclerosis and their diagnostic and therapeutic potential in cardiovascular diseases.

Regarding arrhythmias, circRNAs, such as hsa_circ_0003965, exhibited distinctive expression in AF patients linked to glucagon signaling [80]. Specific circRNAs, such as hsa_circ_0000075 and hsa_circ_0082096, contributed to AF through the TGF-beta pathway [80]. CircCAMTA1, identified as a potential therapeutic target, showed elevated levels in AF tissues and angiotensin-II-treated fibroblasts. Knockdown of circCAMTA1 inhibited fibroblast proliferation and reduced atrial fibrosis-related gene expression, alleviating AF in vivo [81]. CircCAMTA1 suppressed miR-214-3p, indicating its potential as a promising therapeutic target for AF [81].

Circular RNAs (circRNAs) have significant potential in clinical applications for cardiovascular diseases, but there are also some challenges. On the positive side, circRNAs play a crucial role in regulating gene expression, which is beneficial for disease diagnosis, particularly in diabetes, neoplasia, and cardiovascular disorders. Specific circRNAs have emerged as potential biomarkers for conditions such as heart failure and essential hypertension, providing a non-invasive and potentially early diagnostic tool. Moreover, they affect cardiac function, remodeling, and regeneration, with some, including CircPostn, CircFASTKD1, and Circ HIPK3, linked to cardiac repair and regeneration [46,48,49,50,51]. These molecules are also involved in cardiac development, as evidenced by their distinctive expression during differentiation.

However, challenges persist in fully utilizing circRNAs for clinical applications. Further research is needed to fully understand the intricate relationships between circRNAs, miRNAs, and their regulatory mechanisms in cardiovascular diseases. Despite recent advances, the potential effects of circRNAs on pathogenic processes in the cardiovascular system still need to be fully understood, necessitating more in-depth investigations. Developing novel cardiovascular disease diagnosis and treatment approaches based on circRNA knowledge demands substantial effort and time. In conclusion, ongoing research is essential to unlock the full clinical potential of circRNAs and address the existing gaps in understanding.

## 8. Materials and Methods

This paper represents a qualitative systematic review that brought together research on the role of circular RNAs in cardiovascular pathology. We systematically searched for research evidence from primary qualitative studies and drew the findings together.

The research evidence was collected from the PubMed database and Elsevier database. We analyzed a total of 108 research papers, from which we processed the corresponding information of 81 articles. The articles included in the study were scientific papers written in English, published in a relatively recent period, 2012–2023. A few articles published before 2012 were used due to the valuable information they provided. Scientific research that did not fit into the chosen topic or did not present relevant information for the preparation of the current study was excluded.

## 9. Conclusions

In recent years, circRNAs have become a significant research topic due to their wide range of biological functions. CircRNAs can potentially be valuable biomarkers for the diagnosis and prognosis of cardiac diseases. An overview of circRNAs and their role in cardiac disease was presented in this review for scientists and clinicians. It is clear that circRNAs play a significant role in cardiac disease. However, many challenges still need to be addressed before we can fully understand how they contribute to the field. Further research is needed to investigate the potential utility of circRNAs as biomarkers and therapeutic targets for heart disease.

## Figures and Tables

**Figure 2 ijms-25-02986-f002:**
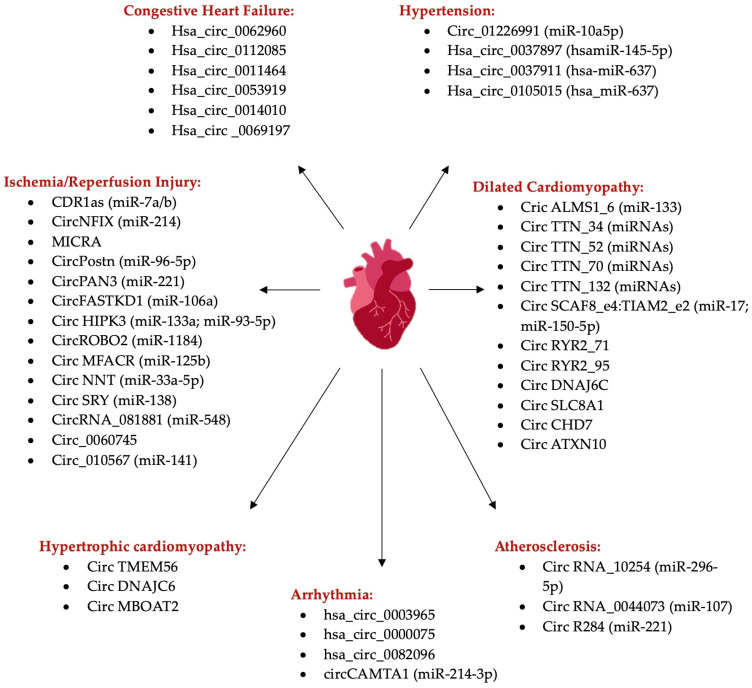
Circular RNAs, their targets (listed between brackets), and their implication in cardiovascular diseases. Created in Notability app.

**Figure 3 ijms-25-02986-f003:**
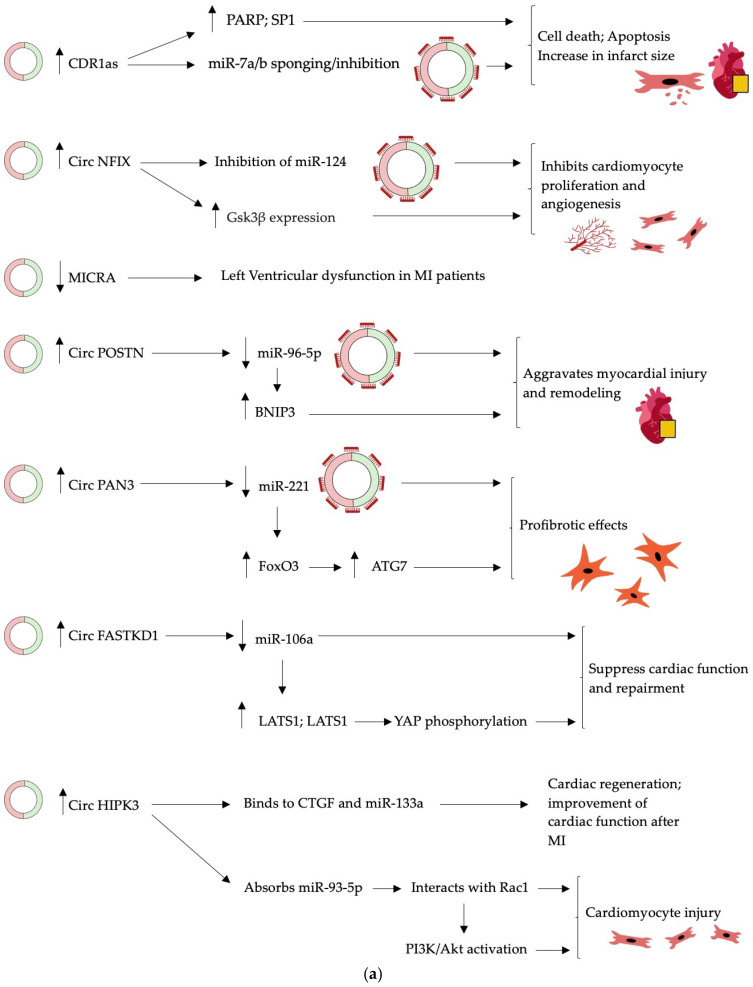
(**a**) Mechanism of action of circular RNAs and their linkage to cardiovascular diseases. (**b**) Mechanism of action of circular RNAs and their linkage to cardiovascular diseases. **↑** = upregulated; **↓** = downregulated. Drawings created in Notability app.

**Table 2 ijms-25-02986-t002:** CircRNAs and EH.

Circ RNA	Target	Effect	Reference
Circ_01226991	miR-10a-5p	Increased risk for EH	[36]
Hsa_circ_0037897	hsamiR-145-5p	Increased risk for EH	[37]
Hsa_circ_0037911	hsa-miR-637	Increased risk for EH, altered serum creatinine/LDL levels	[38]
Hsa_circ_0105015	hsa-miR-637	Increased risk for EH, associated with inflammatory pathways/endothelial dysfunction	[39]

**Table 3 ijms-25-02986-t003:** Circ RNAs, I/R injury, and MI.

Circ RNA	Source	Target	Action Mechanism	Effect	Reference
CDR1as	Mouse heart, cardiomyocytes	miR-7a/b	CDR1as/miR-7	Increasing infarct size; promoting cell death and apoptosis	[15,16,18,42]
CircNFIX	Mouse heart, cardiomyocytes	miR-214	miR-214/Gsk3β	Inhibits cardiomyocyte proliferation and angiogenesis	[43]
MICRA	MI patients	-	-	Low levels were associated with left ventricular dysfunction	[44,45]
CircPostn	MI patients, cardiomyocytes, mouse heart	miR-96-5p	miR-96-5p-BNIP3	Aggravates myocardial injury and remodeling	[46]
CircPAN3	Rat hearts	miR-221	miR-221/FoxO3/ATG7	Profibrotic effects	[47]
CircFASTKD1	HCMEC	miR-106a	miR-106a-LATS1/2-YAP	Suppress cardiac function and repairment	[48]
Circ HIPK3	Mouse heart, cardiomyocytes	miR-133a; miR-93-5p	circHIPK3-miR-133a-CTGF miR-93-5p-Rac1-PI3K/Akt	Cardiac regeneration; Cardiomyocyte injury	[49,50,51]
Circ ROBO2	Mouse heart, cardiomyocytes	miR-1184	miR-1184-TRADD	Promotes cardiomyocyte apoptosis	[52,53]
Circ MFACR	MI patients, cardiomyocytes, mouse hearts	miR-125b	miR-125b	Promotes cardiomyocyte apoptosis	[54,55]
Circ NNT	MI patients, cardiomyocytes	miR-33a-5p	miR-33a-5p-USP46	Promotes pyroptosis and aggravates I/R injury	[56]
Circ SRY	cardiomyocytes	miR-138	miR-138-MLK3/JNK/c-Jun	Reduces apoptosis, regulates Circ NNT and USP46	[57,58]
CircRNA_081881	Plasma of patients with AMI	miR-548	miR-548-PPAR	Potential target for precision diagnosis and treatment	[59]
Circ_0060745	Mouse heart	-	NF-kB	Increasing size of infarct and worsening of cardiac function	[60]
Circ_010567	Rat heart	miR-141	miR-141/DAPK1	Reduced cardiomyocyte apoptosis, suppressed caspase 3 activity	[61,62]

**Table 4 ijms-25-02986-t004:** CircRNA and dilated cardiomyopathy.

CircRNA	Level	Target	Effect	Origin	Reference
Circ ALMS1_6	-	miR-133	Cardiac remodeling	-	[69]
CircTTN_34	Downregulated	miRNAs	Cardiac remodeling	TTN gene	[69]
CircTTN_52	Downregulated	miRNAs	Cardiac remodeling	TTN gene	[69]
CircTTN_70	Downregulated	miRNAs	Cardiac remodeling	TTN gene	[69]
CircTTN_132	Downregulated	miRNAs	Cardiac remodeling	TTN gene	[69]
CircSCAF8_e4: TIAM2_e2	Downregulated	miR-17; mir-150-5p	-	-	[69,70]
circRYR2_71	Downregulated	-	Altered amino acid sequence and disruption of protein structure and function	RYR2 gene	[69]
circRYR2_95	Downregulated	-	Altered amino acid sequence and disruption of protein structure and function	RYR2 gene	[69]
circDNAJ6C	Downregulated	-	-	-	[71]
circSLC8A1	Upregulated	-	-	-	[71]
circCHD7	Upregulated	-	-	-	[71]
circATXN10	Upregulated	-	-	-	[71]

**Table 5 ijms-25-02986-t005:** CircRNA and hypertrophic cardiomyopathy.

Circ RNA	Level	Purpose of Assay	Reference
Circ TMEM56	Downregulated	Quantitative indicators of disease severity	[73]
Circ DNAJC6	Downregulated	Quantitative indicators of disease severity	[73]
CircMBOAT2	Downregulated	-	[73]

**Table 6 ijms-25-02986-t006:** Circ RNA and atherosclerosis.

Circ RNA	Target	Effect	Reference
CircRNA_102541	miR-296-5p	Progression of atherosclerosis	[76]
CircRNA-0044073	miR-107	Proliferation of human vascular smooth muscle cells and human vascular endothelial cells	[77]
Circ R284	miR-221	Proliferation of vascular wall	[78]

**Table 7 ijms-25-02986-t007:** CircRNAs and arrhythmias.

Circ RNA	Action mechanism	Effect	Reference
hsa_circ_0003965	Glucagon signaling pathway	AF	[80]
hsa_circ_0000075	TGF β signaling pathway	AF	[80]
hsa_circ_0082096	TGF β signaling pathway	AF	[80]
circCAMTA1	circCAMTA1/miR-214-3p/TGFBR1	AF; potential therapeutic target	[81]

## Data Availability

Not applicable.

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
