# Peer review of "The Role of Circular RNA for Early Diagnosis and Improved Management of Patients with Cardiovascular Diseases"

_ijms, 2024, doi:10.3390/ijms25052986_

Round 1
Reviewer 1 Report
Comments and Suggestions for Authors
Authors reviewed role of circular RNA is cardiovascular diseases. Overall idea of the study is of interest, however, paper requires corrections.
1. Paper sounds too technical, there are three chapters (1-3) that should be reduced to major information.
2. More clinical observations and conclusions are required. The pros and cons for clinical application of circular RNA should be provided.
3. Any figures demonstrating mechanism of action of circular RNAs and thier linkage with cardiovascular diseases should be presented.
4. If the targets for several circular RNAs were provided, authors could make a next step and analyse role of the predicted targets for cardiovascular diseases. Perhaps, there is a strong chain of relationship between various markers.
Author Response
We express our gratitude to esteemed editors and reviewers for their valuable feedback, which significantly contributed to enhancing the quality of this paper. We have diligently addressed all points in the revised manuscript in response to the comments and suggestions received. We have incorporated changes throughout the document in alignment with the reviewer's recommendations to enhance the clarity of study objectives and results communication. Specific responses to individual comments are outlined below. It is important to note that the page and line numbers mentioned by the reviewers in their feedback may not align with the revised version of the manuscript. Therefore, we have highlighted and cross-referenced the changes according to the new line/page numbers in the revised version for the reviewer's convenience.
- Q (query): Paper sounds too technical, there are three chapters (1-3) that should be reduced to major information.
A (answer): We reduced to major information and rephrased chapters 1-3.
- Q: More clinical observations and conclusions are required. The pros and cons for clinical application of circular RNA should be provided.
A: Lines 356-394; lines 405-451: We added more clinical observations and conclusions. We provided the pros and cons of the clinical application of circular RNA.
- Q: Any figures demonstrating mechanism of action of circular RNAs and thier linkage with cardiovascular diseases should be presented.
A: We created and inserted into manuscript Figures 1, 2, 3a, and 3b for a better presentation. A figure (figure 3a; 3b) demonstrating the mechanism of action of circular RNAs and their linkage with cardiovascular disease was presented.
- Q: If the targets for several circular RNAs were provided, authors could make a next step and analyse role of the predicted targets for cardiovascular diseases. Perhaps, there is a strong chain of relationship between various markers.
A: lines 195-203: We added a paragraph presenting the targets for several circular RNAs in the context of hypertension.
Lines 215-228: We added a paragraph presenting the targets for several circular RNAs in the context of ischemia/ reperfusion injury.
Lines 272-277: We added a paragraph presenting the targets for several circular RNAs in the context of dilated cardiomyopathy.
Lines 301-308: We added a paragraph presenting the targets for several circular RNAs in the context of atherosclerosis.
We also created Figure 2 representing the association between circular RNAs, their targets, and cardiovascular diseases.
Once again, we appreciate your time, expertise, and commitment to maintaining the high standards of our academic community. Your thorough review has undoubtedly played a crucial role in shaping the final version of our paper.
We appreciate your dedication to advancing scientific knowledge and look forward to any additional feedback you may have.
Best regards,
Claudia Goina
On behalf of all authors
Reviewer 2 Report
Comments and Suggestions for Authors
The manuscript presents a nice review of the role of circular RNA in cardiovascular diseases and their diagnostic usefulness, preceded by a general presentation of circular RNA (which is quite appropriate). It is an actual research subject so the review may be of interest to researchers. In my opinion, it is generally well written, having only some flaws of mainly editorial nature.
Lines 38/39: are troponins enzymes?
Table 1. Please provide the title of the last column (References)
Tables 3,4 and 6: please increase the width of the column devoted to “Effects”, if possible
Tables 2-6: please correct “Refference”
Table 5: The term “Purpose” is unclear. I understand that the authors meant purpose or usefulness of the assay
Lines 281-282: “supplies valuable information about disease diagnosis”, perhaps better:… “useful for disease diagnosis” or something similar
Lines 312-315: This paragraph is appropriate rather for a summary than as a part of “Materials and Methods” and can be omitted.
References 5, 17, 27, 28, 31, 32, 33 40, 62 and 69. Please provide full bibliographic data. The same for Ref. 55 if published in the meantime.
Ref. 4, 35. Please standardize the way of citation. Check also for other references.
Comments on the Quality of English LanguageEnglish OK but I found a spelling error "Refferences"; perhaps a check would be useful.
Author Response
We express our gratitude to esteemed editors and reviewers for their valuable feedback, which significantly contributed to enhancing the quality of this paper. We have diligently addressed all points in the revised manuscript in response to the comments and suggestions received. We have incorporated changes throughout the document in alignment with the reviewer's recommendations to enhance the clarity of study objectives and results communication. Specific responses to individual comments are outlined below. It is important to note that the page and line numbers mentioned by the reviewers in their feedback may not align with the revised version of the manuscript. Therefore, we have highlighted and cross-referenced the changes according to the new line/page numbers in the revised version for the reviewer's convenience.
- Q (query): Lines 38/39: are troponins enzymes?
A (answer): Lines 35/36: Troponins, important cardiac enzymes changed to troponins, the most well-known and significant cardiac proteins. Troponins are a complex of regulatory proteins found in skeletal muscle and cardiac muscle.
- Q: Table 1. Please provide the title of the last column (References)
A: Table 1: We added Reference as a title for the last column.
- Q: Tables 3,4 and 6: please increase the width of the column devoted to “Effects”, if possible
A: Table 2,3,4,5,6: We increased the width of all columns to improve the aspect of the tables. We also increased the width of the column devoted to “Effects” for a better view of the text.
- Q: Tables 2-6: please correct “Refference”
A: Tables 2-6: The term “Refference” (incorrect spelling) was corrected to “Reference” (correct spelling).
- Q: Table 5: The term “Purpose” is unclear. I understand that the authors meant purpose or usefulness of the assay
A: Table 5: The term “Purpose” changed to “Purpose of assay”. The corrected form is more transparent and more understandable.
- Q: Lines 281-282: “supplies valuable information about disease diagnosis”, perhaps better:… “useful for disease diagnosis” or something similar
A: Lines 345-346: “Supplies valuable information about disease diagnosis” changed to “provides valuable information regarding disease diagnosis” for a more explicit statement.
- Q: Lines 312-315: This paragraph is appropriate rather for a summary than as a part of “Materials and Methods” and can be omitted.
A: Lines 452-453: We omitted the paragraph because it didn’t meet the criteria for the “Materials and Method” chapter.
- Q: References 5, 17, 27, 28, 31, 32, 33 40, 62 and 69. Please provide full bibliographic data. The same for Ref. 55 if published in the meantime.
A: References 5, 17, 27, 28, 31, 32, 33, 40, 55, 62, 69 We provided complete bibliographic data.
- Q: 4, 35. Please standardize the way of citation. Check also for other references.
A: References 4, 35 We standardized the way of citation. We also standardized the way of citation for other references and provided full bibliographic data.
Once again, we appreciate your time, expertise, and commitment to maintaining the high standards of our academic community. Your thorough review has undoubtedly played a crucial role in shaping the final version of our paper.
We appreciate your dedication to advancing scientific knowledge and look forward to any additional feedback you may have.
Best regards,
Claudia Goina
On behalf of all authors

Reviewer 3 Report
Comments and Suggestions for Authors
The document is a comprehensive review focusing on the role of circular RNAs (circRNAs) in the diagnosis and management of cardiovascular diseases (CVDs). It highlights circRNAs' significant potential as biomarkers and therapeutic targets due to their involvement in gene expression regulation and their presence in various biological processes. The review discusses circRNAs' implications in various cardiovascular conditions, including heart failure, hypertension, ischemia/reperfusion injury, myocardial infarction, cardiomyopathies, and atherosclerosis.
1. Offer a more detailed explanation of the molecular mechanisms by which circRNAs influence cardiovascular pathology.
2. Emphasize the clinical implications of circRNA research, including potential for therapeutic intervention and personalized medicine.
3. Highlight the importance of interdisciplinary collaboration between molecular biologists, cardiologists, and bioinformaticians in advancing circRNA research.
4. Please add the research and discussion about arrhythmia and circRNA.
Author Response
We express our gratitude to esteemed editors and reviewers for their valuable feedback, which significantly contributed to enhancing the quality of this paper. We have diligently addressed all points in the revised manuscript in response to the comments and suggestions received. We have incorporated changes throughout the document in alignment with the reviewer's recommendations to enhance the clarity of study objectives and results communication. Specific responses to individual comments are outlined below. It is important to note that the page and line numbers mentioned by the reviewers in their feedback may not align with the revised version of the manuscript. Therefore, we have highlighted and cross-referenced the changes according to the new line/page numbers in the revised version for the reviewer's convenience.
- Q (query): Offer a more detailed explanation of the molecular mechanisms by which circRNAs influence cardiovascular pathology.
A (answer): We added figures 3a and 3b to explain the molecular mechanisms of circRNAs in cardiovascular pathology.
- Q: Emphasize the clinical implications of circRNA research, including potential for therapeutic intervention and personalized medicine.
A: Lines 135-143: We emphasized the clinical implication of circRNA research, including the potential for therapeutic intervention and personalized medicine
- Q: Highlight the importance of interdisciplinary collaboration between molecular biologists, cardiologists, and bioinformaticians in advancing circRNA research.
A: Lines 51-57: We highlighted the importance of interdisciplinary collaboration between molecular biologists, cardiologists, and bioinformaticians in advancing circRNA research.
- Q: Please add the research and discussion about arrhythmia and circRNA.
A: Lines 310-342: We added a chapter researching arrhythmia and circular RNAs. Lines 425-432: We added a discussion about arrhythmia and circRNAs.
Once again, we appreciate your time, expertise, and commitment to maintaining the high standards of our academic community. Your thorough review has undoubtedly played a crucial role in shaping the final version of our paper.
We appreciate your dedication to advancing scientific knowledge and look forward to any additional feedback you may have.
Best regards,
Claudia Goina
On behalf of all authors

Round 2
Reviewer 1 Report
Comments and Suggestions for Authors
Authors corrected their paper accordingly. It benefits from the revision.